# Topology and Robustness of Weighted Air Transport Networks in Multi-Airport Region

**Bingxue Qian \* and Ning Zhang \***

School of Economics and Management, Beihang University, Beijing 100191, China
\* Correspondence: 18600268389@163.com (B.Q.); nzhang@buaa.edu.cn (N.Z.); Tel.: +86-137-0116-1902 (N.Z.)

**Abstract:** Topological analyses of multi-airport regional air transport networks are the basis for the sustainable development of multi-airport systems. In this study, we modeled the Yangtze River Delta (YRD) region's airport infrastructure as a network and presented a weighted approach by which to analyze the network structure and robustness from the perspective of complex network theory. The analysis of the Yangtze River Delta Airport Network (YRDAN) indicates that it is a small-world network, and its cumulative degree has a power-law distribution, suggesting that it has scale-free properties. As its weighted clustering coefficient was found to be much smaller than the non-weighted counterpart, this demonstrates that most of the network traffic is focused on a hub-and-spoke pattern. Furthermore, the over-centrality of the YRDAN suggests weak accessibility of small cities and high dependence of air transport on the hub-and-spoke pattern. The assessment of the robustness of the YRDAN in the face of intentional attacks found that domestic networks are more robust than foreign aviation networks. However, the isolation of a small fraction of selected nodes can cause serious problems in the functioning of the YRDAN.

**Keywords:** complex network; air network; topology; robustness; multi-airport region

## 1. Introduction

With the rapid development of the civil aviation transport industry in China, the air transport network (ATN) has become one of the most crucial infrastructure networks for the development of the economy, as it strongly promotes the spatial movement of people, goods, capital, and information. According to the International Air Transport Association (IATA) [1], 1352 million passengers were carried by China's air transport networks (CANs) in 2019, which was an increase of 6.9% from 2018. As one of the most economically dynamic and strategic regions of China, the YRD region witnessed powerful growth and had the largest proportion of total CANs. The passengers carried by the YRDAN in 2019 amounted to 266 million, and the growth rate was 16.9% from 2018. The total number of airports in the YRD region is 23, and the airport density has reached 0.8 per square kilometer. IATA also reviewed the fast-growing air transportation market in terms of air passenger volume and reported that the demand grew faster than capacity [2]; this demonstrates the congested network structure of the YRDAN and great heterogeneities in the capacity and strength of connections.

With the development of complex network theory in recent years, there has been an increased interest in the study of ATNs [3–5]. The study of complex networks began by defining new concepts and measures, using them to describe the topology of real networks. The result was the identification of principles and statistical properties of real networks. As the study of complex networks has intensified, the main interest of research has gradually focused on dynamic behavior. Topological analysis, based on complex networks, helps further our understanding of network characteristics and the properties of their dynamic behaviors. This may help in the study of phenomena, such as robustness, resilience, or propagation processes. In order to analyze the topology and robustness of an ATNs, it is

desirable to abstract and integrate its various complex and heterogeneous elements in order to assess its uncertainty and other properties of interest without much detail. Complex network theory provides a theoretical framework that may help the development of appropriate models and analyses of the topology and robustness of ATNs. Based on complex network theory, ATNs can be modeled as a graphical (network) model, with airports as nodes and flights as links, and the topology and robustness of ATNs can be analyzed by proposing a theoretical model. There have been some studies analyzing ATNs at the global, regional, and national levels [6–8], however, this is less in terms of exploration of ATNs from a multi-airport-region perspective. Topology and robustness analyses of a multi-airport region can provide policymakers and related managements with network route planning recommendations from a coordinated development perspective, thereby reducing the impact of disruptions for intentional or unintentional reasons, improving the transport efficiency of air transport, and ensuring that multi-airport regional air transport has good communication, i.e., maximizing the robustness of their networks at a reasonable cost.

The main contributions of this study are presented in two aspects. First, a weighted network statistical analysis was applied to the multi-airport region to characterize the evolving topological structure of the YRDAN over time. Second, an efficiency index weighted by the number of available seats between airports, which has not been widely used in air transportation networks before, was applied to the robustness study of the multi-airport region to expand the robustness study of the related aspects of aviation networks. To this end, the rest of the paper is organized as follows. The literature review regarding the application of complex network theory in the study of air transport networks is presented in the second section. Our methodology and data are discussed in the third section. The results of the complex network and robustness analysis of the YRDAN are discussed and analyzed in the fourth section, followed by an overview of our main findings and the implications of this study in the fifth section.

## 2. Literature Review

The study of complex networks began with the definition of new concepts and measures by which to characterize the topology of real networks. The result was the identification of the principles and statistical properties of real networks [9–11]. There have been a number of applications of complex networks theory in transportation networks, such as roads [12], railways [13], subways [14], and maritime transport [15]. Until recently, it has also been widely used in the field of air transport networks [16–18].

Based on the theory of complex networks, the ATN is modeled as a network graph consisting of airports as nodes and routes between airports as edges. It is obvious that ATNs are neither simply random nor regular. Thus, a complex network analysis of their structures begins with the identification of topological features and patterns. Interestingly, many airport networks present one of two different topological properties: a scale-free property (SF) and a small-world property (SW) [11,19]. These properties have been examined in several studies of ATNs. However, in addition to the abovementioned cross-sectional analysis of ATNs, some other studies have also focused on the historical evolution of ATNs. For instance, the work in [20] discussed the evolution of the Brazilian airport network in the period 1995–2006 and discovered network shrinkages even though the number of passengers and cargo traffic had increased. The experimental analysis of international ATNs conducted by [21], using data from 2002 to 2013, demonstrated that the properties of SF and SW are stable. The work in [22] studied the evolution of Cuba's ATN after a loss of ties with the United States.

However, with the continued expansion of the aviation scale, a dynamic view of its spatial structure needs to be further explored [23]. In a dynamic environment, airports and routes can be temporarily closed for a variety of reasons, such as environmental incidents, security alerts, and strikes or terrorist attacks, resulting in high costs for airlines and countries. For example, the strike by Spanish air traffic controllers caused losses of approximately USD 134 million in 2010 [24]. EasyJet lost up to GBP 31 million due to snow

and strikes in the same year [25]. Flight delays and cancellations in 2012 cost China more than CNY 43.9 billion [26]. Robustness analyses of air transport can estimate the impact of errors (random causes) or attacks (intentional causes) on the route network and can assess the resilience of the network and the tolerance to congestion caused by attacks and malfunctions. Although there is no general definition of robustness in the transportation field, robustness mainly refers to the ability of a system to maintain its performance in the face of disruptions [27]. Therefore, studying the robustness of ATNs is particularly important for improvements in the operational efficiency of route networks [28–30]. Based on conventional topological analyses, several robustness studies have investigated some of the topological metrics that are affected when a portion of the nodes are isolated, such as the clustering coefficient, the average shortest path length, the size of the giant component, and global efficiency [31–33]. These studies evaluate the robustness of the ATN by repeatedly calculating the metrics in different destructive scenarios and assessing the trend of the metrics based on their variability. In addition to conventional topological analyses, robustness analyses based on a multilayer/multilevel analysis have become prevalent [34,35]. These studies decompose the nodes of the ATN and identify the existence of a set of critical nodes in the network. By simulating attacks on the network to detect these nodes, it has been found that the network performance deteriorates dramatically when the critical nodes of the network are isolated [36]. In addition, for random closures and intentional attacks, it has been found that aviation networks have more tolerance for random closures than intentional attacks on a set of critical nodes with a high degree of centrality. Thus, tolerance is defined as the capacity of the system to maintain its connectivity features following random or intentional disruptions to nodes or links [8,37]. These findings highlight the need for regional coordination to mitigate various risks.

Based on the review of the above literature, this paper is led to conclude three main points. First, the analysis of complex network structures has been widely studied in the air transportation field. However, these studies have primarily focused on the global, regional, and national levels; the multi-airport region seems to be under-researched. Second, most studies examined only the static state of the network for one year based on the perspective of complex network theory, with less research on the historical evolutionary state of the network. Third, studies have investigated the robustness of air transport networks facing either closures or target attacks, and efficiency has evolved into a prominent method of assessing network robustness. However, weighted efficiency has not been widely used in the study of air transport networks [33], and weighted networks exhibit considerable heterogeneity in terms of the capacity and strength of connections relative to unweighted networks [23]. Therefore, an analysis using weighted efficiency can help reveal the true nature of the network structure. In this context, this paper examines the evolutionary structure of the YRDAN from 2016 to 2020 based on a weighted complex network analysis, further exploring the robustness of the Yangtze River Delta region using weighted efficiency indicators with a view to contributing to relevant studies on air transportation networks and to provide useful information for network development and planning in other multi-airport regions. In the next section, the methodology and data used in this study are described in detail.

## 3. Methodology

### 3.1. YRDAN Representation

As one of the most economically dynamic and strategic regions of China, the Yangtze River Delta region has a total of 16 airports, including 2 in Shanghai [38]. The airport density of airports within the YRD region has reached 0.8 airports per square kilometer, which means that the air traffic is highly concentrated. In 2019, the airports in the YRD region are connected to 291 airports, allowing air passenger traffic of 266 million passengers, with the total scale of passenger transportation accounting for about 19.67% of the total passenger traffic of domestic airports. In general, airports in multi-airport regions are classified based on a certain distance from the hub [39] or on a legally defined locale in which the

airports are situated. According to [40], here we similarly define airports within a two-hour public transport time as airports within the same MAR. Since Shanghai, Hangzhou, and Nanjing are important hub airports in the Yangtze River Delta region, and taking Shanghai, Hangzhou, and Nanjing as the center, the other airports belonging to the YRD region in this study are Hangzhou International Airport (HGH), Nanjing International Airport (NKG), Ningbo International Airport (NGB), Shuofang International Airport (WUX), Hefei Xinqiao Airport (HFE), Nantong Xingdong Airport (NTG), Yancheng Nanyang Airport (YNZ), Yiwu Airport (YIW), Changzhou Airport (CZX), Shanghai Pudong Airport (PVG), and Shanghai Hongqiao Airport (SHA). It is worth noting, however, that when a city is served by more than one airport, such as Shanghai, all airports in the same city are abstracted as one node and traffic is combined for purposes of comparison.

For the purpose of developing the YRDAN, edges were created between each airport pair if any passenger flight connected these two airports. This study was conducted by modeling, using all the route data from 2016 to 2020 for the YRD multi-airport region. The data used in the study are from the Official Airline Guide (OAG). Therefore, we built the YRDAN model as a connected network $G = (N, E)$, where $N = \{1, 2, 3, \cdots n\}$; $N$ is the number of nodes where airports are abstracted; and $E$ is the number of edges where the flights between airport pairs $(N_i, N_j)$ are abstracted. In complex networks, weighted networks incorporating the continuous nature of the network can reflect the connection strength or link weight [8,41,42], which enables a better understanding of the statistical features of real-world systems [43]. To accommodate the traffic information in the network in this study, the YRDAN was modeled as a weighted undirected network, where the weights are represented by the total number of available seats from airport $N_i$ to airport $N_j$ and defined by the weight matrix $A_W$, that is, $W_{ij}$ represents the weight between the airport pair, which is as follows in Equation (1):

$$A_w = \left[ W_{ij} \right]_{n \times n} \tag{1}$$

### 3.2. Network Structure Measures

Each type of network shows specific topological features characterizing the connectivity, interactions, and dynamic processes within the network [43]. Therefore, the measurement of the most relevant topological features of complex networks facilitates the identification and description of the complex statistical characteristics of the networks [44]. In this study, several metrics were used to measure the topology of the YRDAN, including the degree distribution, average path length, clustering coefficient, and centrality metrics.

### 3.2.1. Node Degree and Distribution

In network theory, $N$ is the total number of nodes that represent the airports of the YRDAN. The number of total connections between node $N_i$ and other nodes in the network are defined as the node degree $K_i$, which is as follows in Equation (2):

$$K_i = \sum_{j \in N} a_{ij} \tag{2}$$

where $a_{ij}$ depends on the presence of flights between airport pairs. If there is a flight between airports, $a_{ij} = 1$; otherwise, $a_{ij} = 0$. The degree distribution, as an important characteristic of networks, is considered a descriptive statistic of the degree of network nodes. If a network has $N$ nodes and $N_K$ of them have degree $K$, the degree distribution $P(K)$ is defined as a fraction of these $K$-degree nodes. The $P(K)$ is shown as follows in Equation (3):

$$P(K) = N(K)/N \tag{3}$$

In the weighted network, the degree of the node $N_i$ is replaced by the strength of the node $S_i$, which indicates the number of flows (operations) linked with the node, as shown as follows in Equation (4):

$$S_i = \sum_{j \in N} a_{ij} W_{ij} \tag{4}$$

### 3.2.2. Average Shortest Path Length

The average shortest path length $L$ is a significant indicator of the performance in the transportation and communication of networks and is defined as the average number of edges along the shortest paths for all possible node pairs in the network [16]. Generally, the smaller the L, the more compact and reachable the network. The $L$ is shown as follows in Equation (5), where $d_{ij}$ is the number of edges for the shortest path from the node $N_i$ to $N_j$.

$$L = \frac{2}{N(N-1)} \sum_{j \in N, i \neq j} d_{ij} \tag{5}$$

### 3.2.3. Clustering Coefficient

The clustering coefficient $C_i$ is defined as the probability that two nodes are connected to each other given that both are connected to node $N_i$, which represents the network's transitivity, written as

$$C_i = \frac{2M_i}{k_i(k_i - 1)} \tag{6}$$

where $M_i$ indicates the actual number of edges between the neighbors of node $N_i$. The average clustering coefficient $C$ is the mean value of $C_i$ of all $N$ nodes in the network, shown as

$$C = \frac{1}{N} \sum_{i=1}^{N} C_i \tag{7}$$

However, to consider the reality of the network, the weighted clustering coefficient $C_i^w$ was proposed to measure local cohesiveness by considering the interaction strength that exists on the local triplet [45]; it is written as follows in Equation (8). Similarly, the weighted clustering coefficient of the network is given by the average of all the $C_i^w$.

$$C_i^w = \frac{1}{S_i(k_i - 1)} \sum_{j,l} \frac{w_{ij} + w_{il}}{2} a_{ij} a_{jl} a_{li} \tag{8}$$

### 3.2.4. Centrality

Centrality metrics are often used to measure the relative importance of network nodes; they include degree centrality, closeness centrality, and betweenness centrality [4]. Degree centrality $C_D(i)$ is the ability of node $N_i$ in the network to directly establish connections with other nodes, which can reflect the importance of node $N_i$ in the network, shown as

$$C_D(i) = k_i/(N-1) \tag{9}$$

For the whole network:

$$C_D = \frac{\sum_i (C_{Dmax} - C_D(i))}{N-2} \tag{10}$$

Closeness centrality $C_c(i)$ is measured by the shortest path between node $N_i$ and other nodes in the network, which reflects its accessibility in a given network, written as

$$C_c(i) = (N-1)/\sum_{j=1, i \neq j}^{n} d_{ij} \tag{11}$$

For the whole network:

$$C_c = \frac{\sum_i (C_{cmax} - C_c(i))}{(N-1)(N-2)} (2N-3) \tag{12}$$

Betweenness centrality $C_B(i)$ is used to measure the extent to which a particular node lies between other nodes in a network, which reflects the control role of the node in the network over the dissemination of information to other nodes, and is shown as

$$C_B(i) = \frac{2\sum_{j \neq i \neq l} N_{jl}(i)/N_{jl}}{(n-1)(n-2)} \tag{13}$$

where $N_{jl}$ is the number of shortest paths from node $l$ to node $j$, and $N_{jl}(i)$ is the number of those paths that pass through node $i$. For the whole network:

$$C_B = \frac{\sum_i (C_{Bmax} - C_B(i))}{N-1} \tag{14}$$

*3.3. Robustness Assessment*

In recent years, the study of ATN robustness has attracted the attention of some scholars. Obviously, the use of different topological metrics will lead to the evaluation of network robustness in different ways. In previous studies, the metrics commonly used to evaluate robustness include the size of the giant component [5,46], the clustering coefficient [34,47], the average shortest path length [4,16], and efficiency [33,48]. The size of the giant component is a representation of the proportion of nodes contained in the largest subset of the network, which is very intuitive and easy to calculate, but it also has the obvious disadvantage of not taking into account the distance between node pairs. The average shortest path length is generally appropriate for applications in well-connected networks. In order to overcome this, the authors of [49] proposed a new metric called efficiency $E$, written as

$$E = \frac{1}{N(N-1)} \sum_{i \neq j \in N} e_{ij} = \frac{1}{N(N-1)} \sum_{i \neq j \in N} \frac{1}{d_{ij}} \tag{15}$$

However, one important feature is ignored when employing efficiency to measure robustness: link weight. It is obvious that the weight between different airport pairs differs with respect to the frequency of flights or the number of seats offered. Therefore, in this paper, weighted efficiency $WE$ is used as a metric of network performance after an attack. By using the number of available seats between airport pairs as weights, it is also easy to find that the weight in the YRDAN is the similarity weight, for which the higher the number of available seats between airport pairs, the stronger the connection between airports. This also means that the higher the weight, the more efficient the airports are to each other. Additionally, efficiency is inversely proportional to distance, which means that for the similarity weights, the weights are inversely proportional to distance. If node $N_j$ and node $N_i$ are directly connected, then $d_{ij}^w = 1/w_{ij}$; otherwise, $d_{ij}^w = \frac{w_{ik} + w_{kj}}{w_{ik} w_{kj}}$, where node $(i,j)$ are connected through node $N_k$. Thus, combining the weights, the efficiency is shown as

$$WE = \frac{1}{N(N-1)} \sum_{i \neq j \in N'} e_{ij} = \frac{1}{N(N-1)} \sum_{i \neq j \in N'} \frac{1}{\sum_{l \in L_{ij}} \frac{1}{wl}} \tag{16}$$

## 4. Results

*4.1. Analysis of Structure*

To explore and analyze the route network development in the YRD multi-airport region from 2016 to 2020, we established a route network map in the region with airports as nodes and flights between airports as sides using the collected data of all flights from

the airports in the nine cities in the region during the period using networks, shown in Figures 1 and 2. In addition, considering the authenticity of the route network, the total number of seats in the airport pair was used as a weight.

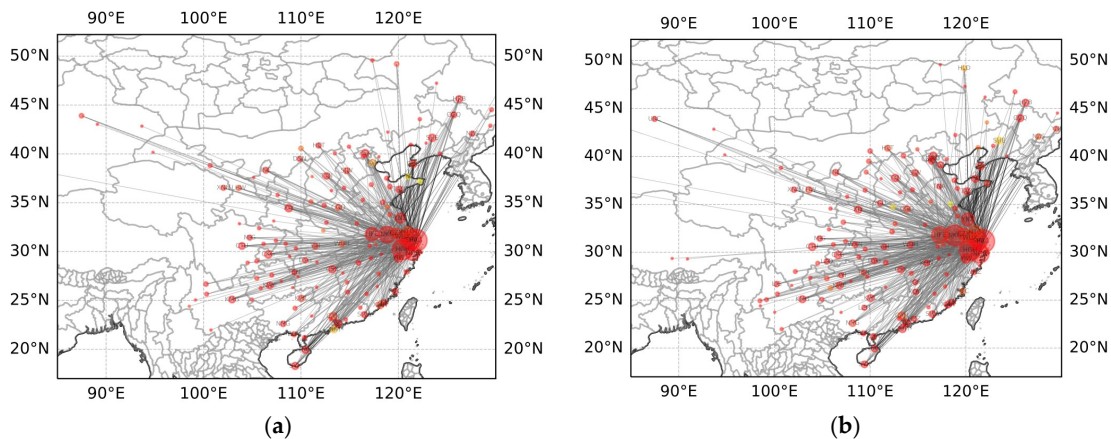

(**a**)　　　　　　　　　　　　　　　　(**b**)

**Figure 1.** Domestic networks of YRDAN: (**a**) 2016, (**b**) 2020.

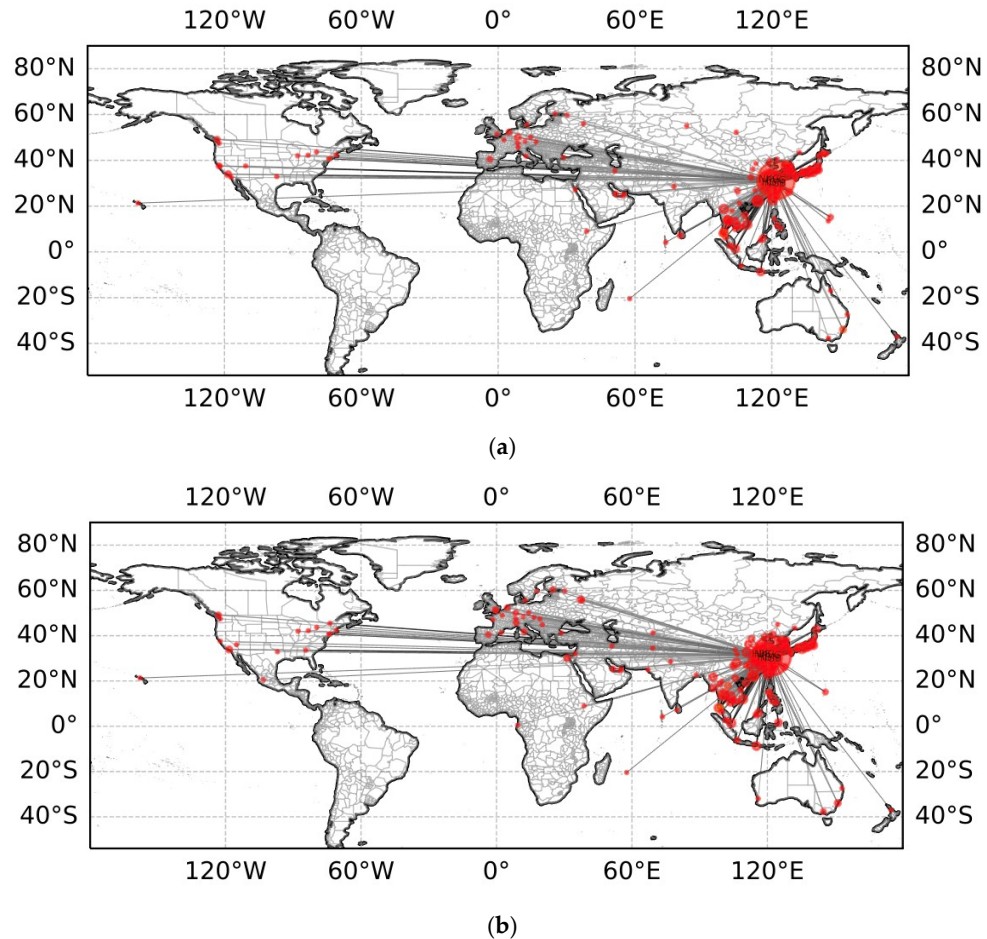

(**a**)

(**b**)

**Figure 2.** International networks of YRDAN: (**a**) 2016, (**b**) 2020.

In terms of the domestic pattern of regional route development for several airports in the YRD region, air passenger traffic was most closely linked to East, South, and North China during this period. Among the YRD region, Shanghai, Nanjing, and Hangzhou were the top cities in terms of frequency and passenger traffic; domestic routes were seen to be

gradually forming a development pattern with Shanghai, Nanjing, and Hangzhou as the center and radiating to other cities.

In terms of the international scale, the international cities of the YRD region are mainly located in Asia, with a few routes connecting Europe, North America, Africa, and Oceania. The overall trend of the international routes in the period from 2016 to 2020 shows a rise. The number of routes between the airports within the YRD region and Europe, North America, and Oceania fluctuates very slightly. The main cause of the network fluctuations is the Asian routes, due to the development of small and medium-sized regional airports, such as Changzhou, Nantong, Yancheng, and Yiwu. In addition, the international airline city of the YRD region is mainly Shanghai, which is the only one that has opened African routes. With the development of Hangzhou and Nanjing airports, these airports established several US and Canadian routes one after another in 2016. Apart from Shanghai, the only airports operating a small number of European routes are Hefei, Hangzhou, and Nanjing airports. It is evident that although international air transport is gradually shifting to cities such as Nanjing and Hangzhou, the development of the international network pattern in the Yangtze River Delta region is extremely uneven.

*4.2. Analysis of Topology*

This paper begins with an analysis of the network structure of the YRNDAN for the average degree and degree distribution in 2020, where the measures of the average degree and degree distribution can provide an overall structural view for the analysis of complex networks [6]. The degree of each airport in the YRD region declines quickly, with the average degree of the YRDAN being 98 and its maximum being 263. The most important Shanghai airport is connected to a majority (about 88%) of all the nodes, while the other airports are connected to 30% of the nodes at most. As can be seen in Figure 3, the cumulative degree distribution follows a power-law distribution: $P(>k) \sim k^{-0.97} (R^2 = 0.95)$, indicating that there are several busy airports in the YRDAN that operate a large number of routes. Therefore, the YRDAN is consistent with the heterogeneity distribution, and it shows scale-free properties within a moderate range of degree values.

Considering the overall picture of network complexity, another important network structure measure is the node's strengths $s(i)$, which describes the total weight of its connections. In order to capitalize on the relationship between the strength and degree of a node, we examined the dependence of $s(k)$ on $k$. It was found that the $s(k)$ of vertices with degree $k$ increases as $s(k) \sim k^{2.05} (R^2 = 0.95)$, shown in Figure 4. It can be seen that well-connected airports can handle more traffic, as expected. The characteristic of the YRDAN is similar to that of the Australian and Indian airport networks [6,16].

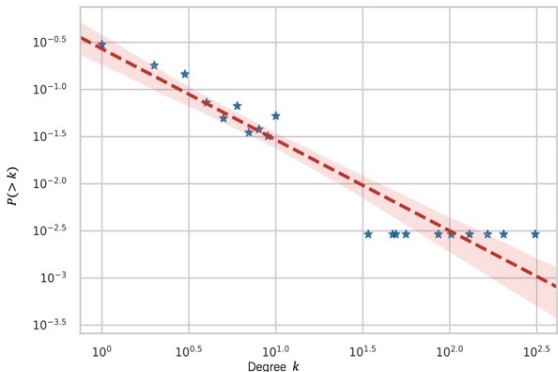

**Figure 3.** Cumulative degree distribution.

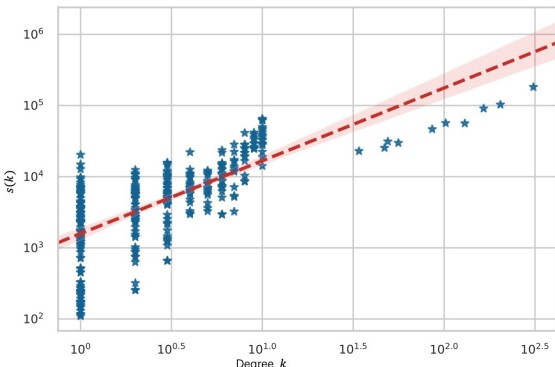

**Figure 4.** Average strength as function of degree.

The average shortest path length is always used to measure the accessibility of traveling in a network. Figure 5 presents the average shortest path length of the YRDAN and a comparable random network of the same size. In the case of the YRDAN, the average shortest path length for these five years is roughly unchanged: all are between 2.1 and 2.2. From a transportation perspective, this implies that a passenger will need one flight change or stopover to reach any city pairs. However, the same-sized random network is much larger than the YRDAN, indicating better air transport convenience in the YRD region.

The clustering coefficient $C_i$ is used to represent the local cohesion to a node. A large-value clustering coefficient indicates a more compact connection system between a node and its neighbors. The average clustering coefficient $C$ measures the overall density of the interconnected nodes in the network to reflect the convenience of network transportation. As shown in Figure 6, the clustering coefficient is continuously significantly higher than that of a comparable random network. This suggests that the YRDAN exhibits a higher degree of concentration than a comparable random network.

From this analysis, it can be inferred that the YRDAN has evolved into having a small-world topology as, according to [11], if $L$ increases almost parallel to $\log(n)$, where $n$ is the number of nodes, the corresponding network can be defined as a small-world network. For the YRDAN, $L = 2.13$ and $\log(n) = 2.47$ for $n = 300$. The average clustering coefficient is $C = 0.21$. All of these network features of the YRDAN confirm that it has similar properties to those of a small-world network during the study period.

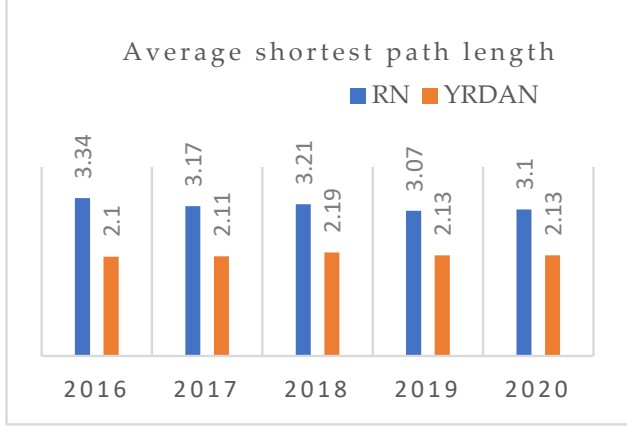

**Figure 5.** Average shortest path length of the YRDAN compared to those of a random network (RN) of the same size, 2016–2020.

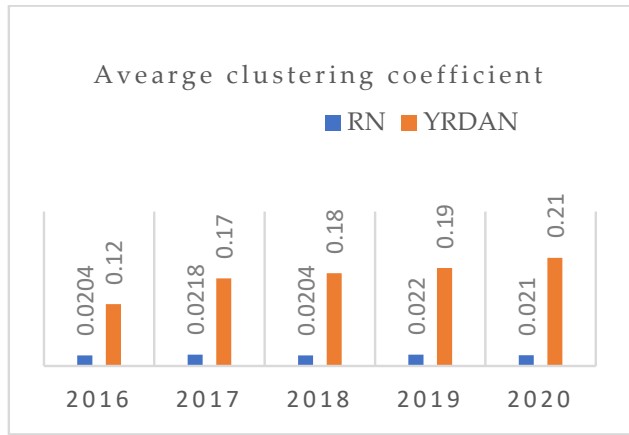

**Figure 6.** Average clustering coefficient of the YRDAN compared to those of the random network (RN) of the same size, 2016–2020.

Since the clustering coefficient does not take into account the interaction strength between airports in the weighted network, in order to solve this problem, we introduced the weighted clustering coefficient $C_i^w$ to combine the weighted information of the network. Let the $C_i^w$ be calculated based on the volumes of traffic operating on the local triplets, which consider the reality of the clustered structure in the network. The average cluster coefficient $C_w$ of the YRDAN is 0.001, which is much smaller compared to its non-weighted coefficient of $C = 0.21$. This shows that the topological clustering is generated by links with low weights, and $C$ has little effect on the organization of the network because the largest interactions of traffic frequency operate on links not belonging to the interconnected triplets. This observation was expected to confirm that a large portion of network traffic is focused on the hub-and-spoke model.

Table 1 compares the topological properties of the YRDAN to those of other similar types of networks. It can be easily observed that the average shortest path of the US network is smaller at 1.93, and the average clustering coefficient is larger at 0.78. The average clustering coefficients of the Indian network, the Australian network, and the Chinese network are also much larger than those of the YRDAN. This means that air transport in the YRDAN is still in an underdeveloped stage, and there is still much room for improving the efficiency of the connection structure of its air transport network.

**Table 1.** Characteristics of air transport networks of the YRDAN and other countries/regions.

| Network | Reference | N | E | L | C | P (>k) |
|---------|-----------|------|--------|-----------|-----------|-------------|
| World | [3] | 3663 | 27,051 | 4.4 | 0.62 | Power law |
| Southeast Asia | [50] | 237 | 602 | 3.12 | 0.21 | Power law |
| US | [50] | 272 | 6566 | 1.84–1.93 | 0.73–0.78 | Power law |
| Italy | [17] | 42 | 310 | 1.98–2.14 | 0.07–0.1 | Pareto |
| India | [16] | 79 | 228 | 2.26 | 0.66 | Power law |
| Australia | [6] | 131 | 596 | 2.9 | 0.5 | Power law |
| China | [4] | 144 | 1018 | 2.23 | 0.69 | Exponential |
| YRD region | | 300 | 973 | 2.2 | 0.21 | Power law |

Table 2 lists the top 10 cities according to degree centrality, closeness centrality, and betweenness centrality in 2020. The ranks for degree and betweenness are generally consistent. As expected, Shanghai is ranked first for degree, closeness, and betweenness, and Hangzhou for degree and betweenness, followed by Shanghai. Nanjing is ranked third for degree and betweenness, and second for closeness. This shows that the YRDAN is extremely dependent on the Shanghai, Hangzhou, and Nanjing airports. However, the value of betweenness in Shanghai is 0.64, which is much larger than that of Hangzhou and Nanjing. This means that Shanghai has a great influence on transfers within the YRDAN,

especially regarding the international network. It is noted that some cities in the YRD region, such as Wuxi, Nantong, Changzhou, and Yiwu, appear in the top 10 cities for degree and betweenness but not for closeness, which shows that these airports in the YRDAN are not well-connected to other airports, and overall accessibility is poor.

**Table 2.** Cities rank by degree centrality, closeness centrality, and betweenness centrality.

| Rank | Degree centrality | Closeness Centrality | Betweenness Centrality |
|------|-------------------|----------------------|------------------------|
| 1 | Shanghai | Shanghai | Shanghai |
| 2 | Hangzhou | Nanjing | Hangzhou |
| 3 | Nanjing | Ningbo | Nanjing |
| 4 | Ningbo | Hefei | Ningbo |
| 5 | Hefei | Hangzhou | Hefei |
| 6 | Wuxi | Yancheng | Wuxi |
| 7 | Nantong | Guangzhou | Nantong |
| 8 | Yancheng | Changchun | Yancheng |
| 9 | Changzhou | Chongqing | Changzhou |
| 10 | Yiwu | Chengdu | Yiwu |

From [51], the overall centrality of a star-shaped network is 100%. This means that the closer the overall centrality is to 1, the more concentrated the network tends to be. For the YRDAN, the overall degree of centrality $C_D = 0.86$; the overall closeness centrality $C_c = 0.77$; and the overall betweenness centrality $C_B = 0.64$. It can be easily observed that the values of both degree centrality and closeness centrality are close to 1, indicating that the YRDAN has a high level of centralization and is at high risk of congestion and attack [52]. Therefore, the balance of the YRDAN needs to be further enhanced to promote its sustainable development.

Figure 7 presents the centrality of the YRDAN and a comparable random network of the same size. In the case of the YRDAN, the centrality values of the nodes fluctuate largely. Conversely, the centrality values of the nodes fluctuate essentially on a horizontal line in random networks. The comparison shows that centrality distribution in the YRDAN is consistent with the SF network characteristics.

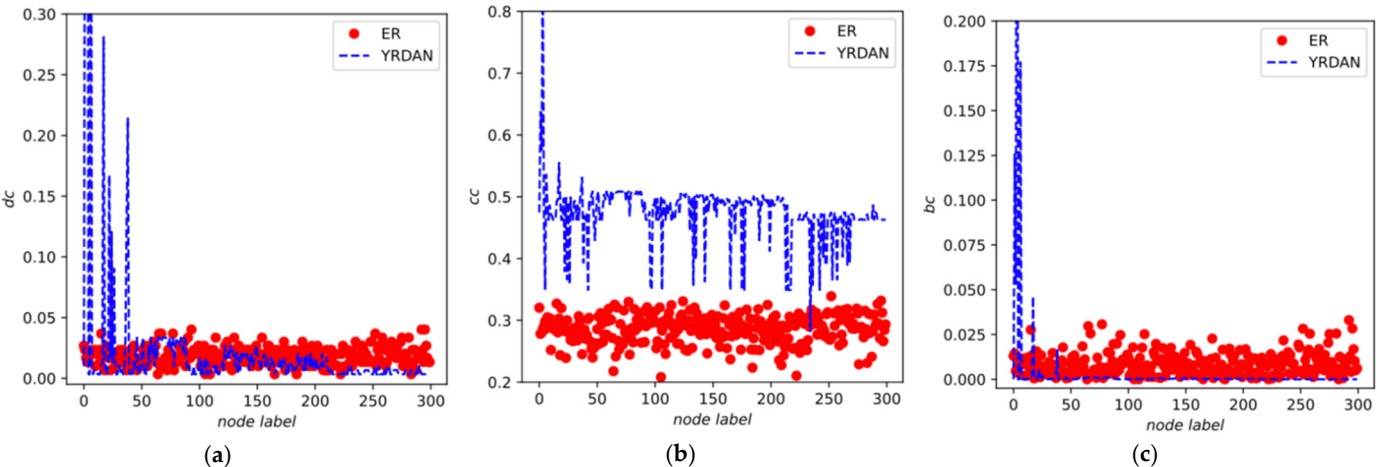

**Figure 7.** Centrality distribution of the YRDAN compared to an RN of the same size. (**a**) Degree centrality. (**b**) Closeness centrality. (**c**) Betweenness centrality.

### 4.3. Analysis of Robustness

The air transportation process may be subject to varying degrees of disruption, such as inclement weather, traffic control, and mechanical failure [53]. It is assumed here that all airports have the same probability of closure [34]. Therefore, to estimate the robustness of random closure, we selected one node at a time as a failed node in the network and its edges

were removed in order to calculate the weighted efficiency of the remaining network. Such a random selection is not related to the topological characteristics or any other attributes of a node.

On the other hand, before the robustness evaluation of an intentional attack on the YRDAN, we identified critical airports in a weighted YRDAN based on the value of degree and closeness. Then, the most important nodes in the network were eliminated, which also led to the failure of the connected edges. Next, the most important nodes in the remaining network were attacked, and the process was repeated until the network was down [33].

Therefore, YRDAN robustness against random closures and attacks can be evaluated by the change in efficiency $WE$ when airports are removed. The robustness of a weighted YRDAN is defined as

$$R_p = \frac{WE_{G_p}}{WE_G} \tag{17}$$

where $WE_{G_p}$ is the weighted efficiency of the remaining YRDAN when airport $p$ and all the links connected to it are removed from the original network $G$. The robustness of the YRDAN against random closures and attacks was assessed and is shown in Figure 8.

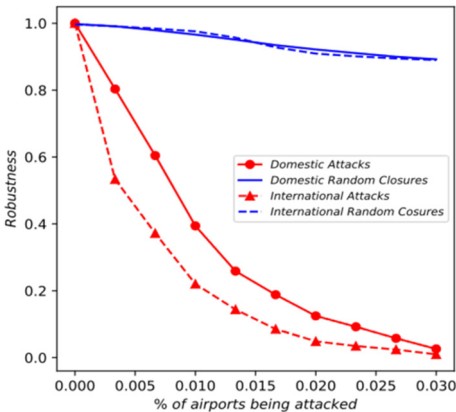

**Figure 8.** Robustness of YRDAN against random closures and attacks.

It can be clearly seen that the robustness values of the YRDAN against random closures and attacks are completely different. Robustness against random closures typically declines in a linear fashion with the proportion of closed airports, while robustness against attacks decreases dramatically when a small proportion of critical airports are attacked. This behavior can be explained by the characteristics of the YRDAN. The topology analysis showed that the YRDAN is an SF network. This is preceded by the means that only a few of these airports have a large number of connections to other airports, while most of them have only a few connections. Hence, the YRDAN has high robustness to random shutdowns and low robustness to attacks, since attacks always target airports with a large number of connections. This shows that the isolation of a small fraction of the selected nodes can cause serious problems to the functioning of the YRDAN, and the overall operation of the YRDAN is extremely dependent on a few core airports.

Furthermore, Figure 8 also shows that the impact of an attack on an international network can be much more significant than the impact of a domestic attack. In the YRDAN, the weighted efficiency values of domestic and international network airports are 0.73 and 0.89, respectively, while simultaneous attacks on domestic and international network airports would result in a decrease in the weighted efficiency values of the network to 80.33% and 53.34%, respectively. Additionally, after three attacks, the weighted efficiency of the domestic network was shown to be 39.41%, and for the international network, it was 22.02%. Therefore, the overall robustness of the YRDAN is poor, while the domestic robustness is higher than that of the international network. It can be seen that the YRDAN is a hub-and-spoke network, and its operations rely on the core airports of Shanghai, Hangzhou, and Nanjing. Compared to the domestic network, Shanghai has an absolute

control role in the international network. Thus, the overall operation of the YRDAN was disrupted when the key node Shanghai was attacked, which means that most of the airports connected to it in the international network were also isolated.

## 5. Conclusions and Implications

### 5.1. Conclusions

In this paper, the evolving YRDAN was examined, and both topology and robustness were analyzed from the perspective of complex network theory. The YRDAN was constructed by associating one node with each airport and linking the different airports using real air traffic data from 2016 to 2020. Additionally, weighted metrics that are quantified by the number of seats offered between airports have also been proposed in this paper. The observations of the topology and robustness of the YRDAN, which may be of practical relevance for policymakers and air service providers, are summarized below.

The topological properties of the YRDAN have exhibited relative stability over the past five years and proved that it has SW characteristics. Moreover, the YRDAN follows a power-law degree distribution, which suggests that it has SF properties.

Compared with random networks, the YRDAN was found to have a larger clustering coefficient and a shorter average shortest path length. However, the YRDAN is at a disadvantage to other similar types of networks, which implies that it remains at a less-developed stage. In addition, the weighted clustering coefficient was found to be much smaller than its non-weighted coefficient, confirming that a large portion of the network traffic is focused on the hub-and-spoke model [6].

The centrality analysis revealed Shanghai's prominent role as a hub airport and the over-centrality of the YRDAN. Additionally, some airports in the YRDAN are not well-connected to other airports, which indicates that partial accessibility is poor. While smaller airports are being integrated into the YRD regional network, connections between these smaller airports and international cities remain virtually non-existent, which again indicates that smaller cities still rely heavily on hub-and-spoke configurations to access the network.

The assessment of the robustness in the YRDAN found that the robustness of the YRDAN to random closures is high, while that attack is low. The isolation of a small fraction of selected nodes can cause serious problems to the functioning of the YRDAN, which can be explained by the SF characteristics. In addition, the overall robustness of the YRDAN is poor and domestic robustness is higher than the international robustness. It can be seen that the development and operation of the YRD multi-airport region are uncoordinated and extremely dependent on a few core airports, especially for international traffic.

### 5.2. Implications

In the context of the coordinated development of multi-airport regions in China, it is particularly important to analyze and plan complex air transport networks in multi-airport regions. Combining the above findings, the following suggestions are made for the planning of the network of the YRD region in order to promote the sustainable development of the YRD region. First, different airports in the region should be positioned differently; the network should be continuously optimized on the basis of strengthening regional coordination in order to reduce the homogeneity of the route network structure between neighboring multi-airport regional airports. Second, to strengthen the robustness and accessibility of the route network within the multi-airport region, more attention should be paid to airports that have strong connections, such as Shanghai airport, Hangzhou airport, and Nanjing airport, while continuously improving the connectivity of the regional route network with important domestic and international hub nodes to ease the operational pressure of these large airports in the region.

There are several potential directions for future research. First, more detailed data can be extended to 20 years. The evolution characteristics of the route network structure in the YRD multi-airport region can be analyzed in a more comprehensive and detailed manner to provide suggestions for the development and planning of route networks in

other multi-airport regions in China. Second, based on the study of the changes in the robustness of the YRD multi-airport region, further studies can be conducted to consider the issue of traffic redistribution, that is, how its passenger flows are transferred to other complementary airports in the region when the airport is closed. Third, in the context of the COVID-19 pandemic, it is worth further study on how to build a relatively robust regional aviation network in the face of the re-emergence of the epidemic, which may cause disruptions to local air passenger networks in several important hub airports within the YRD multi-airport region.

**Author Contributions:** Conceptualization, B.Q.; methodology, B.Q.; software, B.Q.; validation, B.Q.; formal analysis, B.Q.; investigation, B.Q.; resources, B.Q.; data curation, B.Q.; writing—original draft preparation, B.Q.; writing—review and editing, B.Q.; visualization, B.Q.; supervision, N.Z.; project administration, N.Z.; funding acquisition, N.Z. All authors have read and agreed to the published version of the manuscript.

**Funding:** This research received no external funding.

**Institutional Review Board Statement:** Not applicable.

**Informed Consent Statement:** Not applicable.

**Data Availability Statement:** The data presented in this study are available on request from the corresponding author.

**Conflicts of Interest:** The authors declare no conflict of interest.

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
