# Peer review of "Topology and Robustness of Weighted Air Transport Networks in Multi-Airport Region"

_sustainability, doi:10.3390/su14116832_

Round 1

Reviewer 1 Report

The title and abstract reflect the paper's contents well, but it could be helpful to mention that the reference context is the Chinese one.

The document sections are adequately structured.

To improve the paper, some comments are proposed, as follows.

  • some acronyms are missing the first time they were mentioned (e.g., AOG page 4). Please, check them through the entire document;
  • Fig 1, page 4, as it stands, seems useless; it should be zoomed in to show the relevant nodes better;
  • the methodology represents a weak point: it should be enhanced and better explained, also adding a flow chart showing the logical path of the research;
  • as regards table 1 (page 11), it is recommended to add a legend aimed at describing at a glance all the items inserted in the column headings;
  • to make reading more appealing, it would be suggested to summarize specific qualitative/quantitative elements in tables instead of putting them in the text;
  • writing style should be improved, at least by reducing the redundancy of the words;
  • the proposed case study covers a specific context (Yangtze River Delta Airport Network), so it would be helpful to mention a sort of potential transferability of the main concepts (or criteria) to other air network-based environments;
  • Some concepts linking the current airports' status with the covid-19 pandemic should be at least mentioned in the conclusions by thinking to next research steps.

Reviewer 2 Report

Comprehensive, consistent and logical presentation of the material. Nevertheless, there are a few things to fix:

1. The review of literature sources needs to be expanded.

2. It is not recommended to make subsections in the conclusions. Further directions of research should be indicated at the end of the conclusions.

Reviewer 3 Report

The aim of this work is to analyse the topology of the air transport network of a Chinese multi-airport region. The topic is not new, and several paper previously analysed the topology of the air transport network. This paper contributes to existing literature by presenting a case study.

Observations:

Page 1, lines 38-40: “Abstracting and integrating the multiple complex and heterogeneous elements of an airport network to avoid requiring too much detail in the evaluation is the best way to analyze the topology and robustness of an ATN.”: why? What does it mean? Please explain better

Page 2, lines 41-42: “Complex network theory can be used to help to develop appropriate models and analyze the topology and robustness of ATNs by presenting a theoretical research framework.”: this sentence is not clear

Page 2, line 43: “There have been a few studies to analyzing ATN at a global, regional and national level” and also page 3, line 112: “air transportation networks have not been studied enough.”: actually, in my knowledge, many many studies have analysed the topology of air transport networks.

Page 2, lines 47-49: “Topology and robustness analysis of a multi-airport region can provide policy makers and alliance management with route network planning recommendations from the perspective of Coordinate development to improve the transportation efficiency of air transportation. And provide them with advice on substitute plans in the event of an attack on an airport or route”: Please explain better how topological analysis can contribute to the planning process.

Page 2, lines 52-54: “Second, a weighted efficiency index weighted by the number of seats available between airports is proposed to study the robustness of YRDAN.”: is this a new indicator? In my knowledge, this index has been already used (widely) in the literature. Then, you don’t propose a new indicator, you apply it. I suggest modifying the contributions by focusing on the case study. In this perspective, I suggest also modifying the title by focusing more on the case study. Moreover, the title comprehends the word “robustness”, however nothing is said regarding the concept of robustness neither in the introduction and in the literature review. Please add something to introduce, describe, define network robustness.

Page 2, lines 81-83: “Although the locations of the airports are fixed, the routes between them are changing, which exposes them to different types of disruption risks”: it is not clear the relation between complex theory (what has been said before) and disruptions. Why do you introduce the concept of disruption?

Page 3, lines 87-88: “Therefore, it is particularly important that the ATN is designed to enhance its robustness”: as a said before, nothing has been said regarding robustness. Introduce the concept of robustness and provide some literature on it.

Page 3, lines 88-89: “Numerical approaches have been proposed [29, 30] to study tolerance to errors and attacks in complex networks.”: why tolerance to errors? Errors or disruptions? Why is it relevant to introduce the concept of attacks and disruptions in your work?

Page 3, lines 108-111: “However weighted efficiency has not been widely applied in research on air transport networks while weighted analysis can helps reveal the true nature of the network structure and the underlying geographic, political, and economic factors.”: How can weighted efficiency help in revealing political and economic factor?

Page 4, lines 128-130: “In this paper, we define airports in the city as within a two-hour-public transport time to Shanghai, Nanjing, and Hangzhou as air passengers’ target airports within the same multi-airport region [43].”: this sentence is not clear, please explain better

Page 4, line 141: “The data used in the study are from OAG” What is OAG? Please explain better

Figure 1: there are two nodes of a different color (yellow/blue). Why?

Page 5, lines 160-161: “In network theory, N is the total number of nodes that represents airports of the YRDAN”: before you said that the network is composed of V nodes, now N nodes. Symbols are not coherent.

Page 5, equation 1: The definition of the degree is not precise. Is aij equal to 1 if a connection exists between which nodes? What is node j belonging to N ? Is the node Vi, i or j? Symbols are all wrong….

Page 5, lines 164-165: “The degree distribution, as an important characteristic of network nodes, is considered as a descriptive statistic of the degree of network nodes.”: the degree distribution is an important characteristic of the network, not of a node.

Symbols are all wrong, in the entire paper. Check them accurately.

Eq 3: not clear. What do aij and Wij represent?

Page 7, lines 205-207: I think this sentence is out of place

Page 7, line 217: Why “therefore”? It is not clear why you use efficiency.

Page 7, lines 221-223: “It is also easy to find that the weight in YRDAN is similarity weight which higher the weight, the stronger the connection between airports and the smaller the distance. For similarity weight, weights and distances are inversely.” Really not clear…. Moreover, what a “similarity weight” is?

Page 7, lines 229-236: Many parts of this paragraph have been already said previously. Please do not be repetitive (the same can be said for lines 240-248)

Page 7, lines 232-234: “this paper establishes a route network map in the region with airports as nodes and flight between airports as sides by collected all flights data of airports in nine cities in the region during the period using Networks, shown in Fig 2 and Fig 3.”: It is not clear. Moreover, I think that figures 2 and 3 should be explained better.

Page 10, line 282: “Hence, the YRDAN exhibits small-world properties during the study period of this paper”: this should be better explained and motivated.

Page 10, line 293-294: “This observation was expected to confirm that a large portion of network traffic is focused on the hub-and-spoke model.” Please explain better the reasons why.

Page 11, lines 317-319: “It is known that the YRDAN is at a high level of centralization and is at high risk of congestion and attack. Therefore, the balance of the YRDAN needs to be further enhanced to promote its sustainable development.” Where is it known? Please add references. Moreover, the second sentence should be better explained.

Page 12, figure 8: what “ER” in the legend is?

Page 12, par. 4.3. Analysis of robustness: the difference between random attacks and targeted attacks should be explained better.

Page 14, lines 394-395: “Thus, this paper argues that the above findings should be combined with recommendations for sustainable development planning in the YRD region.” Please explain better

Page 14, lines 400-404: “In addition, to strengthen the accessibility and stability of the route network, the connection of the regional route network to important domestic and international hub nodes should be continuously improved. This will increase the accessibility of the regional route network, while relieving the pressure on the operations of large airports in the region.” I do not fully understand the relation between this sentence and the work shown in the paper.

Moreover, the paper is very difficiult to read because of the poor quality of the English. There are MANY MANY MANY mistakes, and the majority of sentences contain grammatical errors. To give a few examples (not exhaustive at all):

Absract: “Furthermore, the over-centrality of YRDAN suggesting weak accessibility of small cities…”; “As its weighted clustering coefficient is much smaller than non-weighted counterpart demonstrates that..”

Introduction: “had the largest proportion of total CANs, Which the passenger carried…” “this demonstrates the YRDAN with congested network structure and great heterogeneities in the capacity and strength of connections.”; “there has been a increase interest…”; “First, the weighted network statistical analysis is applied to the multi-airport region and characterize the evolving topological structure…”

Page 3, from line 123: “In 2019, airports in the YRD region are connected to 291 airports to completed air passenger traffic of 266 million passengers with the total scale of passenger transportation…”

Page 4, line 140-141: “This paper was conducted modeling using all route data”

Page 4, lines 145-146: “In complex networks, weighted networks incorporate the continuous nature of the network can reflect the connection strength or link weight”

Below eq. 1: “Where aij is the flights”

Page 7, lines 221-223 “It is also easy to find that the weight in YRDAN is similarity weight which higher the weight, the stronger the connection between airports and the smaller the distance”

Page 9, line 245: “In terms of international, Shanghai's hub position is more obvious”

Page 10, line 285: “Since the clustering coefficient does not take into account the interaction strength be- 285 tween airports in the weighted network. In order to….”

And MANY, MANY others. Please check accurately.

Round 2

Reviewer 3 Report

Thank you for taking into considerations my observations. In my opinion, the paper has improved. However, there are still some mistakes in the mathemathical formulation and definitions. Symbols continue to be inconsistent, and more than a symbol is used to indicate the same thing. Please, check accurately. 

Equation 1: Specify that weights are indicated with symbol Wij

Page 5, line 186: previously nodes (airports) were indicated with symbols i and j, now with Vi (also at lines 193, 202, 205, 207, 217…). Please use only one symbol

Page 5, line 188: "where ??? is the flights between airport pairs.": this sentence is not completely correct. I suggest changing it: "where aij depends on the presence of flights between airport pairs"

Page 5, lines 190-192: Is node degree K or k? check and use only one symbol

Equation 3: what N(k) and N are?

Line 208: Are nodes in the network V, n or N? Again, use only one symbol (check and correct also successive pages).
